# Application of principal component analysis on temporal evolution of COVID-19

**Ashadun Nobi[1], Kamrul Hasan Tuhin[1], Jae Woo Lee[2]***

**1** Department of Computer Science and Telecommunication Engineering, Noakhali Science and Technology University, Sonapur Noakhali, Bangladesh, **2** Department of Physics, Inha University, Incheon, Republic of Korea

* jaewlee@inha.ac.kr

**Data Availability Statement:** The data underlying the results presented in the study are available from Our World in Data (https://ourworldindata.org/coronavirus-source-data).

## Abstract

The COVID-19 is one of the worst pandemics in modern history. We applied principal component analysis (PCA) to the daily time series of the COVID-19 death cases and confirmed cases for the top 25 countries from April of 2020 to February of 2021. We calculated the eigenvalues and eigenvectors of the cross-correlation matrix of the changes in daily accumulated data over monthly time windows. The largest eigenvalue describes the overall evolution dynamics of the COVID-19 and indicates that evolution was faster in April of 2020 than in any other period. By using the first two PC coefficients, we can identify the group dynamics of the COVID-19 evolution. We observed groups under critical states in the loading plot and found that American and European countries are represented by strong clusters in the loading plot. The first PC plays an important role and the correlations ($C_1$) between the normalized logarithmic changes in deaths or confirmed cases and the first PCs may be used as indicators of different phases of the COVID-19. By varying $C_1$ over time, we identified different phases of the COVID-19 in the analyzed countries over the target time period.

## Introduction

The COVID-19 is an ongoing pandemic caused by the transmission of severe acute respiratory syndrome coronavirus 2 (SARS-CoV-2) [1]. This viral disease began spreading rapidly in Wuhan, Hubei Province, China in December of 2019 [2–4]. The World Health Organization declared COVID-19 as a worldwide pandemic on March 11th, 2020 [5]. At that time, the cumulative number of infected individuals followed a low power growth trend [6]. By maintaining social distance and avoiding human-to-human contact, the spread of the virus can be reduced significantly [7, 8]. However, to limit the power growth of the infection, in addition to practicing social distancing, it is helpful to identify infected individuals and isolate them from others [6].

The governments of various countries have locked down their cities and to ensure proper social distancing among people. Different models have been applied to the time series of COVID-19 confirmed, death and recovered cases to observe and predict the COVID dynamics in different countries [9–18]. To understand COVID-19 dynamics, the susceptible exposed infectious removed model and bi-furcation analysis have been performed [9]. Fractional-order

**Funding:** This work was supported by the Ministry of Education of the Republic of Korea and the National Research Foundation of Korea (NRF-2019S1A5C2A03081234). The funders had no role in study design, data collection and analysis, decision to publish, or preparation of the manuscript.

**Competing interests:** The authors have declared that no competing interests exist.

model is applied to test data which give a close forecasting to the real data [10]. Autoregressive model is applied on real world time series data to forecast the confirmed and recovered COVID-19 cases [11]. The dynamic evolution of COVID-19 in different countries is analyzed using network topology [12]. The maximum time span of the infection and annihilation of the diseases is predicted using susceptible, infectible, quarantine and confirmed recovered model based on only known data of confirmed and recovered cases [13]. Some studies have tried to combine similar countries into groups based on COVID-19 time series data by applying various clustering methods and principal component analysis (PCA) for better realization of its behavior [14–21].

Recently, factor analysis (FA) technique is used to categorize seven severely affected countries in basis of the cumulative counts of cases and deaths due to COVID-19 [14]. The relations between the spread rates of COVID-19 in high risk countries are determined by first Two PCs [15]. The first four principal components of PCA is used to investigate the cyclical patterns of the COVID-19 in different countries [16]. PCA is applied on data of 213 COVID-19 patients and identified three distinct groups of the COVID-19 patients [17]. Principal component analysis based unsupervised feature extraction is applied to RNA expression profiles of 16 COVID-19 patients and 18 healthy control subjects to identify genes associated with COVID-19 [18]. Autoregressive time series models based on two-piece scale mixture is applied on world time series of the confirmed and recovered cases of COVID-19 and forecast the confirmed and recovered cases of COVID-19 [19]. The time series model based on two-piece scale mixture normal distribution is used to predict the number of confirmed cases and death rate of COVID-19 in the world [20]. Fuzzy clustering method is applied on time series of COVID-19 in the high risk countries [21] and clustered the countries considering population's size. Different methods were evaluated for forecasting the trajectories of COVID-19 using its previous data which are useful to determine the estimated number of infected individuals and number of deaths in advance [22–25].

In this article, we apply PCA to the correlations for the changes of cumulative time series of COVID-19 death and confirmed cases between different countries and characterize the evolving correlation structures within countries. By studying the temporal evolution of the correlations of the different affected countries, we uncover periods when there were major changes in the correlation structure of the global pandemic in the analyzed countries. We use the largest eigenvalues of the correlation matrices to observe the COVID-19 dynamics over time. Using loading plots of first two PC coefficients onto two dimensional spaces, we find different groups of the analyzed countries over time following the state of pandemic. We also find that countries make strong cluster when pandemic state is severe among the countries. Finally, the significant changes of diseases that occurred different countries are identified from the change of first PC coefficients.

In this paper, we adopt a different approach to extract interesting and significant patterns from time series of COVID-19 in the affected countries. We measure the correlations from the fluctuations of COVID-19 cases and deaths among the top 25 countries such that one can imagine how fast the virus has been transmitted in the analyzed countries over the periods. To do so, we calculate the correlations for the logarithmic change of cumulative cases or deaths among the analyzed countries. To understand the correlations in more detail, we perform an eigenvalue analysis of the correlation matrix. We also perform eigenvectors analysis corresponding to first two eigenvalues. We calculate the PC coefficients which are correlations between PCs and normalized change of death or confirmed cases. The group dynamics of the affected countries is observed over time using first two PC coefficients not done before.

## Methods

Let us define the logarithmic change in mortality caused by COVID-19 in country $i$ as

$$d'_i(t) = \ln[N_i(t)] - \ln[N_i(t-1)], \tag{1}$$

where $N_i(t)$ is the cumulative number of deaths caused by COVID-19 in a country on day $t$. The normalized change in mortality for country $i$ is defined as

$$d_i(t) = (d'_i(t) - <d'_i>)/\sigma_i, \tag{2}$$

where $\sigma_i$ is the standard deviation of the death in time series $i$ over the time window and the symbol $<\cdots>$ denotes average over the time window. The size of the time window is a month. Next, we construct a normalized death matrix $D$ for 25 countries with dimensions of $N \times T$. The correlations of the changes in deaths among countries can be calculated as

$$C = \frac{1}{T}DD^T, \tag{3}$$

where $D^T$ is the transpose of $D$. The correlation matrix $C$ can be diagonalized in the following form:

$$C = VPV^T, \tag{4}$$

where $P$ is the diagonal matrix with eigenvalues $\lambda_i = (\lambda_1, \lambda_2, \cdots, \lambda_N)$ in descending order and $V$ is an orthogonal matrix of the corresponding eigenvectors. Each eigenvalue and the corresponding eigenvector can be written as

$$\lambda_i = v_i C v_i^T = v_i Cov(D_t) v_i^T = Var(v_i^T D_t) = Var(y_{i,t}), \tag{5}$$

where $y_{i,t} = v_i^T D_t$ is known as $i_{\text{th}}$ principal component (PC) [26, 27]. The eigenvalues represent the variances of the data in the directions of the eigenvectors. A similar procedure was also performed for COVID-19 confirmed cases to calculate eigenvalues, eigenvectors, and PCs.

### Data analysis

We analyzed the time series of daily COVID-19 cumulative confirmed cases and deaths from April of 2020 to February of 2021 in one month time windows. Data were collected from [28]. The daily logarithmic changes in COVID-19 death cases and confirmed cases were determined and PCA was applied to the correlation matrix of changes in COVID-19 death cases and confirmed cases for the top 25 countries. The countries are 1. United States (US), 2. India (IND), 3. Brazil (BRA), 4. United Kingdom (UK), 5. Russia (RUS), 6. France (FRA), 7. Spain (SPA), 8. Italy (ITA), 9. Turkey (TUR), 10. Germany (GER), 11. Colombia (COL), 12. Argentina (ARG), 13. Mexico (MEX), 14. Poland (POL), 15. Iran (IRN), 16. South Africa (SAF), 17. Ukraine (UKR), 18. Indonesia (INDO), 19. Peru (PER), 20. Czechia (CZE), 21. The Netherlands (NETH), 22. Canada (CAN), 23. Chile (CHI), 24. Portugal (POR), and 25. Romania (ROMA).

### Eigenvalues and eigenvectors

We now describe the eigenvalues and corresponding eigenvectors of the correlation matrices of COVID-19 confirmed cases and deaths. We analyzed only the largest and second-largest eigenvalues and their corresponding eigenvectors. The time evolution of the first eigenvalue follows evolutionary dynamics because it exhibits sharp changes in some periods. The first eigenvalues represent the largest variances in the data, which can be used to identify severe effects of COVID-19. The dynamical changes in the first and second eigenvalues of COVID-19

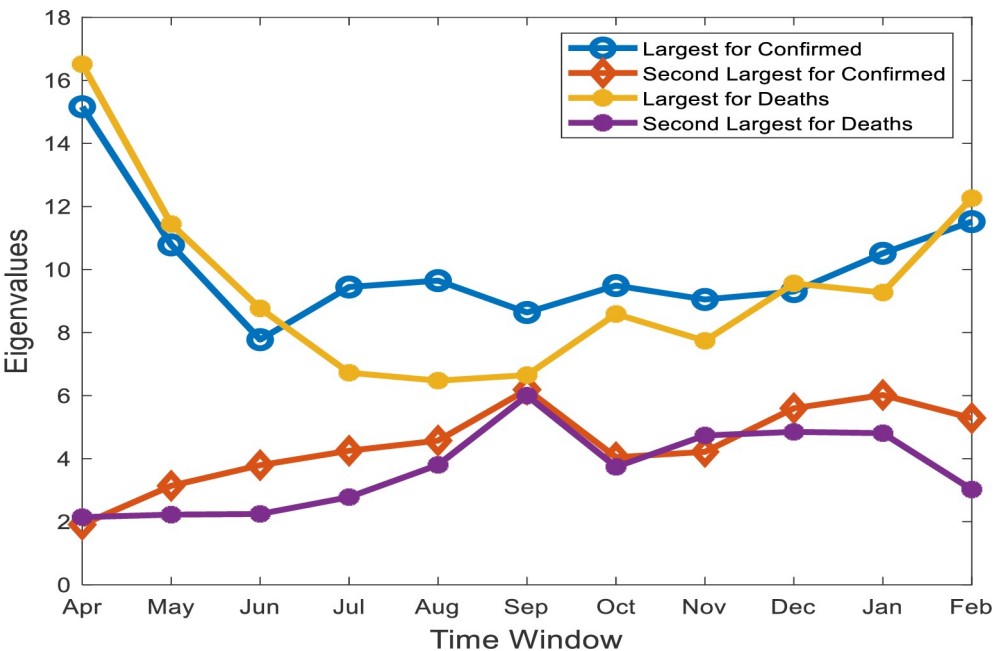

**Fig 1. Changes in the first and second eigenvalues of COVID-19 deaths and confirmed cases.** The largest variances in the data can be observed at the beginning of COVID-19 during April and May.

confirmed cases and deaths over a one month time window are presented in Fig 1. The higher value of the largest eigenvalues suggests that affected countries are more similar in the state of the diseases. In Fig 1, one can see higher values for the first eigenvalues at the beginning of the COVID-19 outbreak, when the virus spread across the world during April of 2020. The largest eigenvalues for confirmed cases begin to decrease until June. Starting in July 2020, there is an increasing trend with small fluctuation for confirmed cases. However, the first eigenvalues for deaths are decreased up to August and then increases with fluctuation till February, 2021 which implies that deaths due to the effect of this disease is raised and correlated between the countries.

When comparing the first eigenvalues of confirmed cases and deaths, one can see that at the beginning of COVID-19 from April to June of 2020, the COVID-19 dynamics of deaths are stronger than those of confirmed cases. After this period, the largest eigenvalue of deaths is lower than that of confirmed cases from July to November and the first eigenvalues are very similar in December. A sharp change in the COVID-19 dynamics of the deaths is observed again in February of 2021, resulting in a crossover with the COVID-19 confirmed cases. This indicates that COVID-19 death cases are more severe in this month.

The second largest eigenvalues for deaths and confirmed cases increase from April to September and are approximately equal to the first eigenvalue of deaths in September. Later, both eigenvalues decrease in October and then increase again starting in November. However, for deaths, it becomes steady till January. When comparing the second eigenvalue of COVID-19 confirmed cases to deaths, one can see that the value of confirmed cases is greater than that that of deaths for most of the target period.

We now consider the components of the first and second eigenvectors of the correlation matrix of COVID-19 confirmed cases and deaths. The components of the first eigenvector corresponding to the largest eigenvalue of the top 25 severe countries over time are shown in Fig 2(A). We found that the components of the eigenvector correspond to the largest eigenvalue

and second largest eigenvalue change sign with evolution of time. We took the absolute value of eigenvectors, which is shown by heat map, because we want how much the components of eigenvector of these countries dominate on the first and second eigenvector. We are not concerned about the sign ambiguity of the eigenvectors. The components of the first eigenvector of COVID-19 confirmed cases and deaths are non-uniform in the target countries. The significant components of this eigenvector for deaths are greater than those of confirmed cases. These components fluctuate significantly with the intensity of diseases and their trends over time are dissimilar between countries. This is reasonable because the severity of diseases do not peak at the same time in different countries.

The most influential components of the first eigenvector of deaths appear for US, Brazil, Russia, UK, Italy, Mexico, and Iran in different periods from June to November, as indicated by the maroon shaded area in Fig 2(A) and 2(C). The least influential components appear for the bottom ten countries, as indicated by the blue shaded areas. The components of the first eigenvector for US deaths are significant in June, August, November, and January, whereas for Brazil, these components are dominant from June to August. The components of the eigenvectors for confirmed cases and deaths in India are the most significant from August to October. The dominance of the eigenvectors for UK and Russia can be observed in different periods from June to November, whereas for Mexico, the greater values appear in July and August.

The most influential components of the first eigenvector for Italy appear from September to January. It is noteworthy that there are no significant components of the first eigenvectors for France and Spain, despite COVID-19 causing many deaths in these countries. The most influential components of first eigenvector for COVID-19 confirmed cases can be observed for most countries in different periods from May to April shown in Fig 2(C). The components of the eigenvector corresponding to the second largest eigenvalue are presented in Fig 2(B) and 2 (D). One can see that the components of the eigenvector of the second eigenvalue of COVID-19 confirmed cases and deaths are much greater that the components of the first eigenvector, but only contribute significantly in few periods, as indicated by the yellow and red shaded areas. The second eigenvector of deaths is more significant than that of confirmed cases. The components of deaths in the US and South Africa are the most dominant in October and April, respectively. Significant components can be identified for most countries in different periods for both confirmed cases and deaths. The most influential components of confirmed cases can be observed for the countries of Brazil, France, Spain, South Africa, Ukraine, Czech Republic, and Portugal. However, other countries also contribute moderately to the second eigenvector in different periods.

## Group dynamics of COVID-19 countries

We now consider the group dynamics on the plane of the first PC (PC1) and the second PC (PC2) as shown in Fig 3. The correlations between PCs and normalized change of death or confirmed cases known as PC coefficients. We use the PC coefficients to measure the strengths of the countries and PCs relationship and to determine which country contributes more on two PCs. The more the contribution of a country on PCs, the more the severity of the diseases is. We projected the first two PC coefficients into two-dimensional spaces to see where the countries are located, which should provide additional information regarding the severity of diseases. The maximum variances of the data in the direction of the largest eigenvector are defined by first PC, while the second PC represents the variances in the direction of the second eigenvector. We selected the first two PCs because the variances explained by the first two PCs are significant for evaluating the severity of COVID-19. Loading plots of the first two PC coefficients for a one month time window are presented in Fig 3.

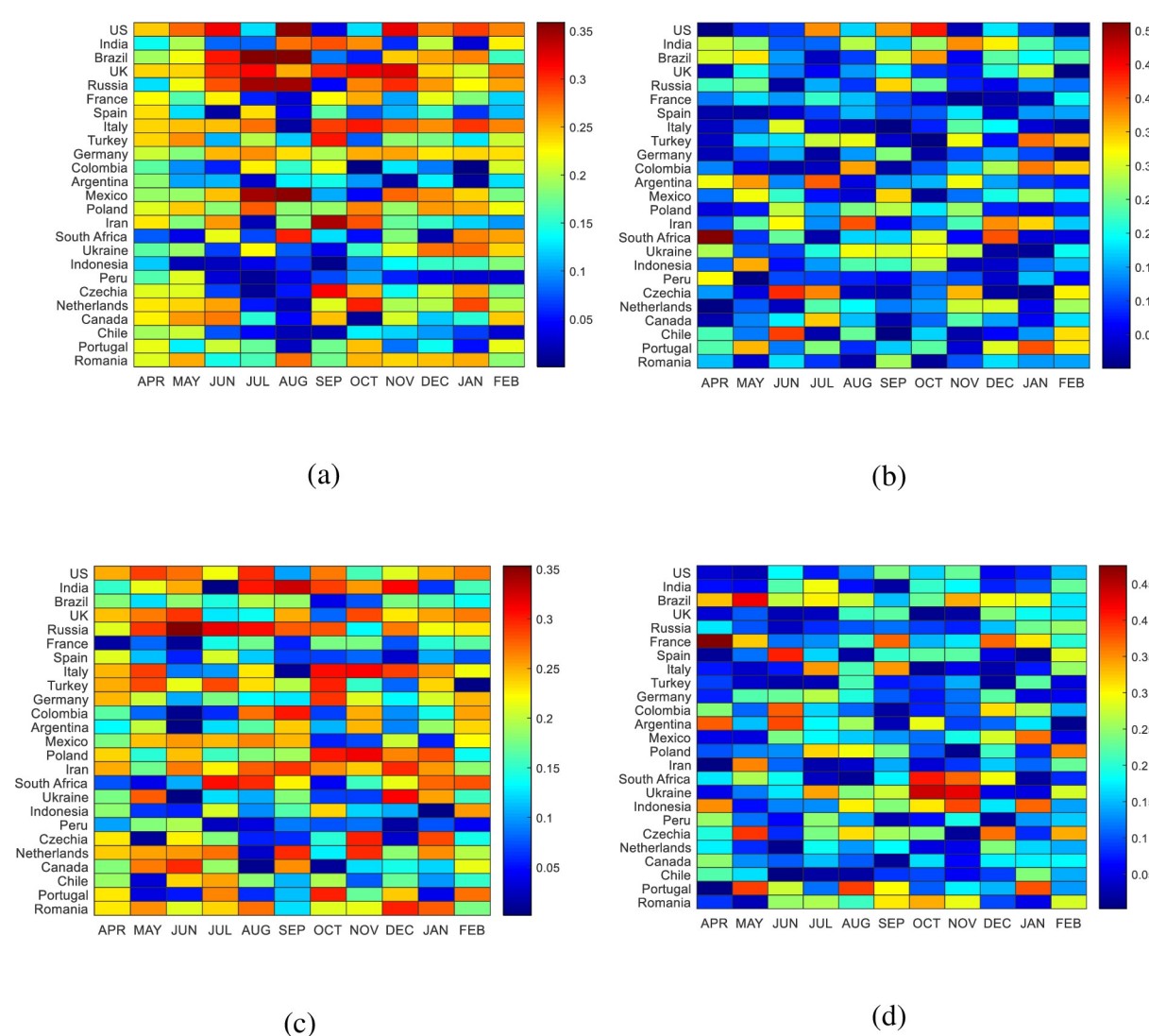

**Fig 2.** Components of eigenvectors (a) the first eigenvector for death cases, (b) second eigenvector for death cases, (c) the first eigenvector for confirmed cases, and (d) the second eigenvector for confirmed cases. The horizontal axis shows the month of the last data point of each time window. The vertical axis represents the affected countries. The maroon shaded areas indicate dominating eigenvectors of different affected countries.

Based on the closest points in the two-dimensional load plots, we identified different groups of countries over time. We observe that changes in the time window altered the groups and clusters. This could be a result of the varying severities of the disease in different countries. We observed that the numbers of clusters and countries contained in a cluster for COVID-19 confirmed cases were not similar to those for COVID-19 deaths. We will explain the group dynamics starting from April of 2020, when the virus began to spread across the world. Almost all countries are in the same cluster in April for confirmed cases. However, when we look for the clusters of COVID-19 deaths, there are two big groups. The cluster that is far from the origin and near to each other indicated by dotted ellipse labeled as 'critical group' in Fig 3(A) contains the countries of the US, UK, Italy, Span, France, Turkey, Germany, Iran, Canada, Czech Republic, Netherland, and Poland. Most of these countries are American and European. This

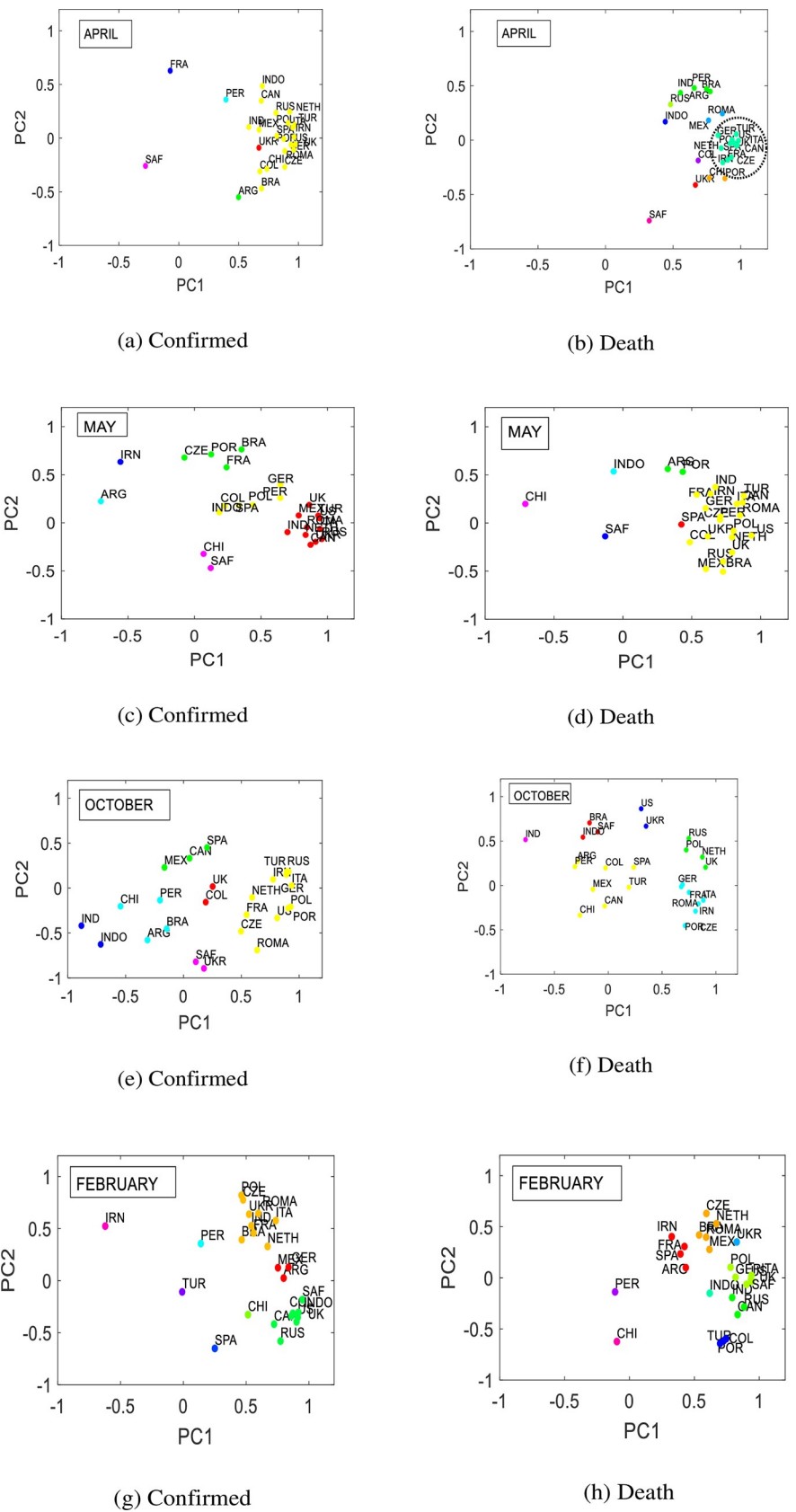

(a) Confirmed

(b) Death

(c) Confirmed

(d) Death

(e) Confirmed

(f) Death

(g) Confirmed

(h) Death

**Fig 3.** Scatter plots of PC1 and PC2 coefficients of the top 25 countries affected by COVID-19 (a) during April 2020 for confirmed cases, (b) April 2020 for death, (c) May 2020 for confirmed cases, (d) May 2020 for death, (e) October 2020 for confirmed cases, (f) October 2020 for death, (g) February for confirmed cases, and (h) February for deaths. We showed only the periods where group dynamic show significant meaning. Countries make strong cluster in the loading plots when the states of the diseases are similar and severe among the countries.

indicates that these countries make the largest contributions to the two PCs. This also indicates that there were critical conditions for these countries caused by COVID-19 in April of 2020.

The countries in the second-largest cluster are Brazil, Argentina, India, Peru, and Russia. The conditions in these countries were moderate during April. The other countries are widely scattered. Regarding the largest clusters in May for COVID-19 confirm cases, three big clusters can be observed, where the largest cluster contains the countries of the US, Canada, Mexico, UK, Netherland, Italy, Romania, Turkey, India. These countries are far from the origin and close to each other, implying a similar state of diseases because they contribute equally to the two PCs. However, the other two clusters are closer to the origin and far from each other, implying the smaller contribution to the two PCs which may be caused by a moderate state of COVID-19 confirmed cases. However, one can see a different scenario for COVID-19 deaths in this month. Excluding Argentina, Chile, Portugal, Indonesia, South Africa, all countries are in the same cluster and we can conclude that the COVID-19 death scenario is almost the same in these countries.

In October, more than ten countries form one big cluster for COVID-19 confirmed cases and there are also other small clusters. However, the points are far from each other overall, indicating varying states of COVID-19. The critical groups split into two and three large clusters in October for COVID-19 confirmed cases and deaths, as shown in Fig 3(E) and 3(F), respectively. The Countries in the large two clusters during October for confirmed cases and deaths are European, implying that the European countries are in a similar situation. The American countries are in the other largest cluster for death group.

Finally, in February, the countries form three and four large clusters for COVID-19 confirmed cases and deaths, respectively, and one can see that the countries of some groups are near to each other and far from the origin, indicating that some countries faced severe diseases scenarios. The group that is far from the origin for COVID-19 confirm cases contains the countries of the US, UK, Canada, Russia, Colombia, Portugal, Indonesia, South Africa, which represent the worst scenarios for COVID-19. Mexico, Germany, and Argentina form another cluster and there is another large cluster shown in Fig 3(G). For COVID-19 deaths, there are two critical groups and the points are close to each other, containing the countries of America, Asia and Europe shown in Fig 3(H). During this period, Iran, France, Span and Argentina form a group near to origin, implying that the contribution of these countries to the two PCs represent good scenarios of COVID-19.

Almost all countries contribute equally to the first two PCs at the beginning of diseases progression and form a strong cluster. However, countries form different clusters over time based on the intensification of diseases because some countries reduce disease severity through lockdown or other countermeasures, resulting in various clusters. Countries that are far from origin and close to each other imply equal contributions to the two PCs and we identify these countries as critical groups. Countries in critical groups are heavily affected by the intensification of diseases. Therefore, we can identify the countries in which COVID-19 is more severe.

## Pandemic states of countries

We now consider the correlation $C_1(d_1, y_1)$ between the normalized logarithmic changes of death or confirmed cases and PC1. The change of correlations $C_1$ over time can be used to identify the COVID-19 states of different countries. Because the largest eigenvalues represent

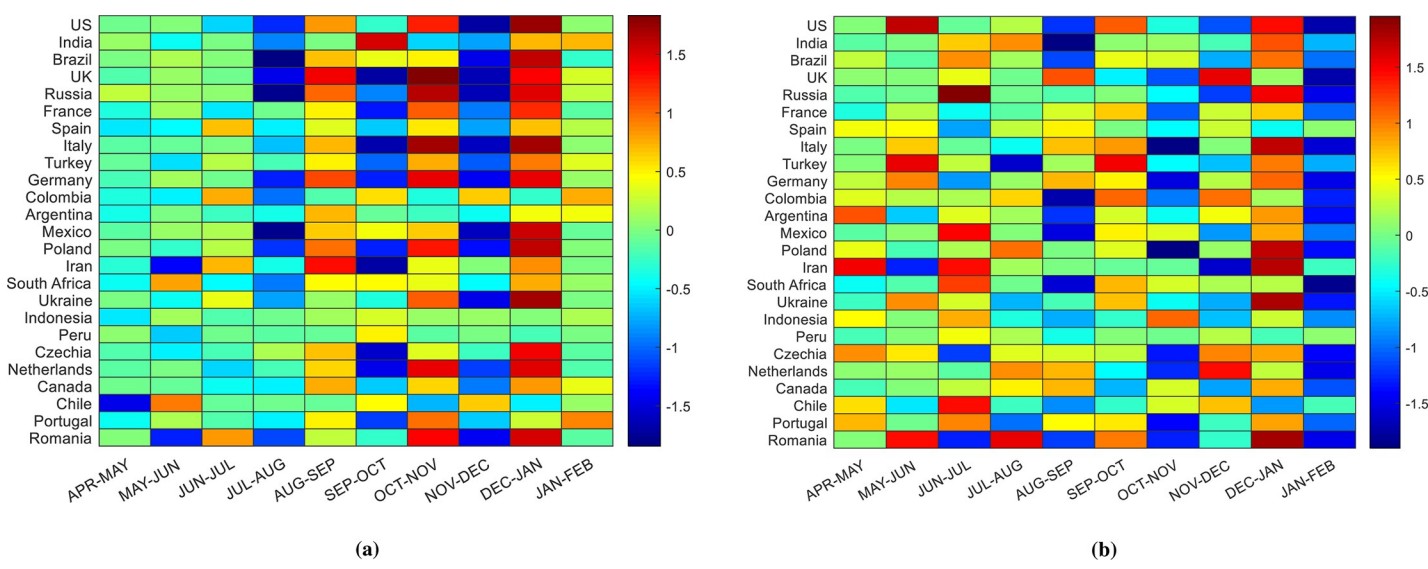

**Fig 4.** Time varying change of PC for (a) COVID-19 death cases and (b) confirmed cases. The change of color indicates the change of the states of the diseases.

the strongest COVID-19 dynamics, the principal PCs denoted by $y_1$ are appropriate for observing COVID-19 states over time. The greatest variations in the data are represented by the first PCs. The temporal changes in the first PCs are also used in finance to identify the states of different markets [17, 18]. We identified the COVID-19 states of countries based on the changes in the correlations of each country contributing to PC1 between two monthly time windows such as $\Delta C_1 = C_1(T+1) - C_1(T)$ where T is the time period. The changes in $\Delta C_1$ for COVID-19 confirmed cases and deaths are visualized by the heat map shown in Fig 4. Changes in color indicate changes in the state of COVID-19 in a country.

Severe states of COVID-19 are identified by red shaded areas, while deep blue shaded areas represent moderate states. Minor changes in COVID-19 state are identified by the yellow color. Excluding the countries of Iran, Chile, and Romania, we found that there were no changes in the states of COVID-19 deaths from April to July in the top countries. A significant change in the COVID-19 state can be observed in August for most countries, implying a moderate state of COVID-19 during this period. In September, severe state of COVID-19 can be observed in some countries such as the UK, Russia, Germany, and Iran. This pattern changes in October excluding the state in India. Most countries enter a severe state of COVID-19 again in November and return to normal state in December. Severe states of COVID-19 deaths can be observed again in January and February in most countries. During these periods, a milder state of COVID-19 death can be observed in Spain, Argentina, Indonesia, Peru, and Chile.

Large changes of COVID-19 confirmed cases cannot be observed for all countries in same periods, as shown in Fig 4(B), which may be a result of lockdowns. Lockdown policies vary from country to country. A significant change in the first PCs can be observed for the US and Russia in June and July, respectively. Some other countries such as Iran, Mexico, Argentina also exhibit state changes from April to July. In September, the mild states of some countries can be observed in Fig 4(B), as indicated by the yellow shaded areas. Small changes in COVID-19 confirmed cases can be observed in most of the countries from September to December, with a severe state returning in December in almost all countries. However, unlike COVID-19 death cases, a moderate state of COVID-19 confirm cases can be observed in February in almost all countries. Additional research is required to analyze this phenomenon, which we will leave for future investigations.

The changes in states of COVID-19 deaths and confirmed cases did not occur for all countries in the same period. In particular, changes in the state of COVID-19 confirm cases can be observed for various countries in different periods from April to December. This may be a result of lockdown policies. However, the changes in states in most countries follow a similar pattern from December to February.

Time series of COVID-19 is an experimental data and researchers have been used different methods to extract information from the data. PCA is an important technique to find out the relation among the affected countries and to give forecast for the transmission of diseases near future. Mahmoudi *et al.*, classify seven severely affected countries into two classes using two factors of FA technique projected onto two dimensional spaces [14]. They also showed the relations between the high risk countries projecting first two PCs onto two dimensional spaces [15]. Using first four PCs, Duarate *et al.*, showed cyclical patterns of COVID-19 over time for various countries. The authors also showed that the incidence curves of the COVID-19 had strong similarities among countries and can predict the trend of cycles near future. Our study finds that the largest eigenvalues of the correlation matrix in the monthly time frame can be used to observe the dynamics of COVID-19 across the world. Since the time evolution of the largest eigenvalue shows the sharp change in a particular period. For example, the sharp rise of the largest eigenvalue in January for confirmed cases imply that the COVID-19 can be worsen near future. The eigenvectors corresponding to the largest eigenvalue are an indication of changes in the correlation matrix. The components of the eigenvector for the largest eigenvalue show non-uniform composition, which imply that some countries contribute it significantly. The non-uniform composition of the components of the first eigenvector can classify the countries in the basis of the severity of the diseases. For example, the significant components of the first eigenvector in August are found for US, Brazil, Russia, Mexico which indicate critical period of the disease in these countries. Moreover, the graphical approach in Fig 2(A) indicate that the components of the first eigenvector are changing over time which identify different periods of the diseases. For example, in case of Brazil, the severe state of the disease was found from June to August while normal state was in September and October. Similarly, we can identify different states of the diseases for other countries using the components of the first eigenvector. We also classify the countries over time using the first two PCs coefficients and observe the similarities of the countries. The time evolution of COVID-19 can be helpful to understand the states and dynamics of the pandemic and to make policy for the future.

## Conclusion

The technique of principal component analysis were used to investigate the time evolving correlation structures of the top 25 affected countries and to study the states of the diseases in these countries. We observed that the diseases dynamics were faster in the beginning of COVID-19 outbreak. During these early periods, COVID-19 deaths were increased faster than confirmed cases. The dynamics of deaths again increased from September with fluctuation and reached at peak in February 2021. The components of the second eigenvector were more significant than the components of the first eigenvector for some countries in some periods.

We also studied the time evolving relationship between different countries by investigating the correlations between the cumulative change of confirmed and deaths by first two PCs. We projected the first two PC coefficients onto two dimensional spaces and found strong clusters in particular periods indicating similar state of the diseases in these countries. The groups that were far from the origin and contained points close to each other can be considered similar and also severe state of the diseases. The first PC contained the largest variances of the data and the correlation $C_1$ between the cumulative change of confirmed and deaths with first PCs

is considered to identify the COVID states of the different countries. Using time varying $C_1$, we identify the periods of a country when it enters in severe state of the diseases and come back to normal state. When we look for the change of $C_1$ for death cases, we found that American and European countries were in severe state of the diseases from September to November 2020. We found that the states of COVID-19 confirmed cases are different than those of COVID-19 deaths and that countries change their states over time. We seem that PCA on time evolving correlations for the change of cumulative COVID-19 cases of the affected countries all over the world can be used to understand the states and dynamics of COVID-19.

We can propose the similar strategic policies to overcome the spreading disease such as the pharmaceutical and nonpharmaceutical methods. Because the dynamics of the spreading disease is changing over time, the PCA depends on the analyzing period. Therefore, the strategic policies are also changing over time. In further work, we will apply feature ranking of machine learning on time series data to find out the relation among the countries and to predict the state of the diseases. More details, we will leave for our next article.

## Author Contributions

**Conceptualization:** Ashadun Nobi, Jae Woo Lee.

**Data curation:** Ashadun Nobi.

**Formal analysis:** Ashadun Nobi, Kamrul Hasan Tuhin.

**Funding acquisition:** Jae Woo Lee.

**Investigation:** Kamrul Hasan Tuhin.

**Methodology:** Ashadun Nobi, Kamrul Hasan Tuhin.

**Project administration:** Ashadun Nobi.

**Software:** Kamrul Hasan Tuhin.

**Supervision:** Jae Woo Lee.

**Validation:** Ashadun Nobi.

**Visualization:** Kamrul Hasan Tuhin.

**Writing – original draft:** Ashadun Nobi, Kamrul Hasan Tuhin, Jae Woo Lee.

**Writing – review & editing:** Ashadun Nobi, Jae Woo Lee.

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
