## [Decision Letter · Decision Letter 0]

20 Sep 2021

PONE-D-21-24857Application of principal component analysis on temporal evolution of COVID-19PLOS ONE

Dear Dr. Lee,

Thank you for submitting your manuscript to PLOS ONE. After careful consideration, we feel that it has merit but does not fully meet PLOS ONE’s publication criteria as it currently stands. Therefore, we invite you to submit a revised version of the manuscript that addresses the points raised during the review process.The manuscript requires further several revisions regarding study novelty and contribution to the related domain, related literature, quantitative framework and discussion, as well as future research directions.

We look forward to receiving your revised manuscript.

Kind regards,

Stefan Cristian Gherghina, PhD. Habil.

Academic Editor

PLOS ONE

Whilst you may use any professional scientific editing service of your choice, PLOS has partnered with both American Journal Experts (AJE) and Editage to provide discounted services to PLOS authors. Both organizations have experience helping authors meet PLOS guidelines and can provide language editing, translation, manuscript formatting, and figure formatting to ensure your manuscript meets our submission guidelines. To take advantage of our partnership with AJE, visit the AJE website (http://aje.com/go/plos) for a 15% discount off AJE services. To take advantage of our partnership with Editage, visit the Editage website (www.editage.com) and enter referral code PLOSEDIT for a 15% discount off Editage services.  If the PLOS editorial team finds any language issues in text that either AJE or Editage has edited, the service provider will re-edit the text for free.

“This work was supported by the Ministry of Education of the Republic of Korea and the National Research Foundation of Korea (NRF-2019S1A5C2A03081234).”

Reviewers' comments:

Reviewer's Responses to Questions

**Comments to the Author**

1. Is the manuscript technically sound, and do the data support the conclusions?

Reviewer #1: Yes

Reviewer #2: Yes

Reviewer #3: No

2. Has the statistical analysis been performed appropriately and rigorously? 

Reviewer #1: Yes

Reviewer #2: Yes

Reviewer #3: Yes

3. Have the authors made all data underlying the findings in their manuscript fully available?

Reviewer #1: Yes

Reviewer #2: Yes

Reviewer #3: Yes

4. Is the manuscript presented in an intelligible fashion and written in standard English?

Reviewer #1: Yes

Reviewer #2: Yes

Reviewer #3: No

5. Review Comments to the Author

Reviewer #1: This is an interesting contribution to the existing literature, but the paper suffers from several shortcomings listed in the following comments.

- The paper should be checked by a native.

- A discussion section should be added.

- The introduction should be updated by recent researches.

- The novelty and contribution should be clearly bolded.

- The authors should consider some works about Data Analysis that can be applied to model different datasets. For example,

- The authors should consider some works about Data Analysis that can be applied to model Covid Datasets. For example,

Time series modelling to forecast the confirmed and recovered cases of COVID-19, Travel medicine and infectious disease 37, 101742.

Modeling and forecasting the spread and death rate of coronavirus (COVID-19) in the world using time series models, Chaos, Solitons & Fractals 140, 110151.

Principal component analysis to study the relations between the spread rates of COVID-19 in high risks countries, Alexandria Engineering Journal 60 (1), 457-464.

Fuzzy clustering method to compare the spread rate of Covid-19 in the high risks countries, Chaos, Solitons & Fractals 140, 110230.

Factor analysis approach to classify COVID-19 datasets in several regions, Results in Physics 25, 104071.

It’s better to suggest some subjects for future works.

Best regards,

Reviewer #2: Outstanding work. PCA in the context of time series is a reliable method to identify similarities between countries in terms of severity and to anticipate new surges. I would suggest to contrast their findings with similar studies in the discussion and to suggest ways to use their analysis in a practical way to forecast new surges and outbreaks. I suggest the authors consider contrasting the study with one produced by our group: Duarte P, Riveros E. Understanding the cycles of COVID-19 incidence: Principal Component Analysis and interaction of biological and socio-economic factors. Annals of Medicine and Surgery. 2021 Jun 1:102437.

Reviewer #3: I failed to identify some points as follows:

1. What is the hypothesis and problem that you address in this paper?

2. Does only PCA enough to explain this complex issue as the scientist are striving to know the behavior of the virus and its mutation.

3. What could be the implications of your study in epidemiology?

4. Is that only focus to check the applicability of PCA methods?

5. `By studying the temporal evolution

of the correlations of the different affected countries, we uncover the periods when there were major

changes in the correlation structure of the global pandemic in the analyzed countries.` to know this country wise data is enough I think what new things can be added by this research? is unclear.

6. The variation in confirmed cases and deaths among different countries is obvious it need not to say after PCA.

7. Why the dynamic trend is changing is not explored here. Multiple factors might be in fact certainly involved from governance to economy.

Overall, I did not find any novel outcome from this study. Thus, I am not recommending its publication in this journal.

6. PLOS authors have the option to publish the peer review history of their article (what does this mean?). If published, this will include your full peer review and any attached files.

Reviewer #1: No

Reviewer #2: No

Reviewer #3: No

---

## [Author Response · Author response to Decision Letter 0]

21 Oct 2021

Thank you for your fruitful comments. 

Reviewer #1: This is an interesting contribution to the existing literature, but the paper suffers from several shortcomings listed in the following comments.

- The paper should be checked by a native.

Ans: A native person checked the paper.

- A discussion section should be added.

Ans: We added the discussion in the manuscript. Please see the last paragraph of result analysis.

- The introduction should be updated by recent researches.

Ans: The introduction is updated by recent articles. We added some sentences in the third paragraph of Introduction part.

- The novelty and contribution should be clearly bolded.

Ans: The novelty is given in the last paragraph of introduction part.

- The authors should consider some works about Data Analysis that can be applied to model different datasets.

Ans: We add the related articles and update our work. We add new references [14-21].

Reviewer #2: Outstanding work. PCA in the context of time series is a reliable method to identify similarities between countries in terms of severity and to anticipate new surges. I would suggest to contrast their findings with similar studies in the discussion and to suggest ways to use their analysis in a practical way to forecast new surges and outbreaks. I suggest the authors consider contrasting the study with one produced by our group: Duarte P, Riveros E. Understanding the cycles of COVID-19 incidence: Principal Component Analysis and interaction of biological and socio-economic factors. Annals of Medicine and Surgery. 2021 Jun 1:102437.

Ans: We added the discussion in the last paragraph of result analysis. We add the reference in [16].

Reviewer #3: I failed to identify some points as follows:

1. What is the hypothesis and problem that you address in this paper?

Ans: We analyze the time series of COVID-19 to extract information about relations and the dynamics of COVID-19 in the top 25 countries. We want to investigate the dynamical similarity among countries. The similarity of strategic policies should be influenced to control the COVID-19. We gave some points in last paragraph of Introduction.

2. Does only PCA enough to explain this complex issue as the scientist are striving to know the behavior of the virus and its mutation.

Ans: Of course, only PCA is not enough to explain these issues. However, we believe that the PCA is one of the powerful methods to understand the dynamics of disease spreading. Genome sequence analysis is necessary to know the behavior of the virus and its mutation. However, genome sequence analysis is critical in understanding the COVID-19 and preventing the spread of the disease. Different techniques such as PCA, Machine learning, artificial neural network can be applied to analyze genome sequence of COVID-19. We will apply these techniques on gene data for further investigation. We include some methods to explain the dynamical features of COVID-19 in second paragraph of Introduction part.

3. What could be the implications of your study in epidemiology?

Ans: The transmission of diseases in the severe affected countries can be understood by using PCA. From the PCA, we can observe the strong correlation of the dynamical change for the spreading disease. We can propose the similar strategic policies to those countries. I include these points in last paragraph in Conclusion part

4. Is that only focus to check the applicability of PCA methods?

Ans: Actually, we apply PCA technique to the correlations for the change of COVID-19 cases and death among the severe affected countries to understand the states and dynamic of COVID-19 in the analyzed countries. This analysis is also helpful to determine the relationships among these countries. In future, we will apply feature ranking of machine learning on gene data to understand the behavior of virus and its mutation.

5. `By studying the temporal evolution

of the correlations of the different affected countries, we uncover the periods when there were major

changes in the correlation structure of the global pandemic in the analyzed countries.` to know this country wise data is enough I think what new things can be added by this research? is unclear.

Ans: PCA is a technique to reduce data dimension. Applying PCA on correlation structure, we try to show that only first two PCs are enough to observe the changes of the states in the analyzed countries and to make relationships among these countries. We gave some comments in the last paragraph in section <Pandemic states of countries>.

6. The variation in confirmed cases and deaths among different countries is obvious it need not to say after PCA.

Ans: Yes. You are right. We wanted to identify the periods to observe the COVID-19 states in the analyzed countries by heat map so that we can make a comparative study among the countries in the basis of severity. 

7. Why the dynamic trend is changing is not explored here. Multiple factors might be in fact certainly involved from governance to economy.

Ans: The change of correlation structure among the countries is identified by the largest eigenvalue and is used to observe the dynamics of COVID-19. The correlation structure for the change of COVID-19 cases and deaths among the countries is altered due to lock down, making social distance, hard immunity and for other policies and consequently, the dynamic trend is changed. Some articles considered other factors like socio-economic factors in new added Ref [16]. In the future study, we will consider other possible factors.

---

## [Decision Letter · Decision Letter 1]

19 Nov 2021

Application of principal component analysis on temporal evolution of COVID-19

PONE-D-21-24857R1

Dear Dr. Lee,

We’re pleased to inform you that your manuscript has been judged scientifically suitable for publication and will be formally accepted for publication once it meets all outstanding technical requirements.

Kind regards,

Stefan Cristian Gherghina, PhD. Habil.

Academic Editor

PLOS ONE

Additional Editor Comments (optional):

Reviewers' comments:

Reviewer's Responses to Questions

**Comments to the Author**

1. If the authors have adequately addressed your comments raised in a previous round of review and you feel that this manuscript is now acceptable for publication, you may indicate that here to bypass the “Comments to the Author” section, enter your conflict of interest statement in the “Confidential to Editor” section, and submit your "Accept" recommendation.

Reviewer #1: All comments have been addressed

2. Is the manuscript technically sound, and do the data support the conclusions?

Reviewer #1: Yes

3. Has the statistical analysis been performed appropriately and rigorously? 

Reviewer #1: Yes

4. Have the authors made all data underlying the findings in their manuscript fully available?

Reviewer #1: Yes

5. Is the manuscript presented in an intelligible fashion and written in standard English?

Reviewer #1: Yes

6. Review Comments to the Author

Reviewer #1: (No Response)

7. PLOS authors have the option to publish the peer review history of their article (what does this mean?). If published, this will include your full peer review and any attached files.

Reviewer #1: No

---

## [Editor Report · Acceptance letter]

23 Nov 2021

PONE-D-21-24857R1 

Application of principal component analysis on temporal evolution of COVID-19 

Dear Dr. Lee:

I'm pleased to inform you that your manuscript has been deemed suitable for publication in PLOS ONE. Congratulations! Your manuscript is now with our production department. 

Kind regards, 

on behalf of

Dr. Stefan Cristian Gherghina 

Academic Editor

PLOS ONE